# High Throughput Preparation of Ag-Zn Alloy Thin Films for the Electrocatalytic Reduction of CO_2_ to CO

**DOI:** 10.3390/ma15196892

**Published:** 2022-10-04

**Authors:** Jiameng Sun, Bin Yu, Xuejiao Yan, Jianfeng Wang, Fuquan Tan, Wanfeng Yang, Guanhua Cheng, Zhonghua Zhang

**Affiliations:** 1Key Laboratory for Liquid-Solid Structural Evolution and Processing of Materials (Ministry of Education), School of Materials Science and Engineering, Shandong University, Jinan 250061, China; 2Taian Institute of Supervision & Inspection on Product Quality, Taian 271000, China

**Keywords:** high throughput, electrocatalysis, CO_2_ reduction, magnetron sputtering, silver-zinc alloy

## Abstract

Ag-Zn alloys are identified as highly active and selective electrocatalysts for CO_2_ reduction reaction (CO_2_RR), while how the phase composition of the alloy affects the catalytic performances has not been systematically studied yet. In this study, we fabricated a series of Ag-Zn alloy catalysts by magnetron co-sputtering and further explored their activity and selectivity towards CO_2_ electroreduction in an aqueous KHCO_3_ electrolyte. The different Ag-Zn alloys involve one or more phases of Ag, AgZn, Ag_5_Zn_8_, AgZn_3,_ and Zn. For all the catalysts, CO is the main product, likely due to the weak CO binding energy on the catalyst surface. The Ag_5_Zn_8_ and AgZn_3_ catalysts show a higher CO selectivity than that of pure Zn due to the synergistic effect of Ag and Zn, while the pure Ag catalyst exhibits the highest CO selectivity. Zn alloying improves the catalytic activity and reaction kinetics of CO_2_RR, and the AgZn_3_ catalyst shows the highest apparent electrocatalytic activity. This work found that the activity and selectivity of CO_2_RR are highly dependent on the element concentrations and phase compositions, which is inspiring to explore Ag-Zn alloy catalysts with promising CO_2_RR properties.

## 1. Introduction

With the worldwide consumption of fossil fuels, the increasing emission of carbon dioxide (CO_2_) in the atmosphere is becoming a more and more serious environmental threat [1,2,3]. Artificial conversion of CO_2_ into fuels and chemical feedstocks is essential to reduce the concentration of CO_2_ and mitigate the greenhouse effect. The available methods include electrochemical, photoelectrochemical, thermochemical, and biological conversions, of which electrochemical reduction is regarded as a clean and effective method due to its mild reaction conditions and ability to use renewable electricity [4,5,6].

Electrochemical CO_2_ reduction reaction (CO_2_RR) can yield a variety of products through two-, four-, six-, and eight-electron pathways, such as carbon monoxide (CO), methane (CH_4_), formic acid (HCOOH), methanol (CH_3_OH) and C2/C3 compounds [3,7,8,9]. CO is the simplest and one of the most common products of CO_2_RR due to the sluggish kinetics of C–C coupling, which can be further electrocatalytically reduced to high-value chemicals or used as a component of syngas to produce hydrocarbon fuels (e.g., aldehydes, methanol, and phosgene) by the Fischer-Tropsch process [10,11]. Therefore, it is important to study efficient catalysts to selectively reduce CO_2_ to CO (${\mathrm{CO}}_{2}{+2H}^{+}{+2e}^{-}\;↔{\;\mathrm{CO}+H}_{2}O$) [12]. However, the first step of CO_2_RR requires a larger overpotential compared to hydrogen evolution reaction (HER: ${2H}^{+}{+2e}^{-}\;↔{\;H}_{2}$), which will inevitably occur as a main competitive reaction in the aqueous CO_2_ reduction system [13,14]. The challenges of CO_2_RR are the high overpotential, kinetically sluggish multi-electron transfer process, and the selectivity against the HER. Developing promising CO_2_RR catalysts with high activity and selectivity will be the key point to efficiently utilizing CO_2_.

Various metals (e.g., Au, Ag, Zn) have been identified as potential electrocatalysts to overcome the sluggish kinetics of CO_2_ activation and improve the selectivity of CO [12,15,16]. The Sabatier principle shows that adsorbed CO (CO*) is an important reaction intermediate for the production of hydrocarbons and oxygen-containing compounds, and optimal binding energy of CO* (E_B_[CO]) can reduce the yield of other products and lead to higher selectivity of CO [17]. Au and Ag show a high selectivity of CO in CO_2_RR, due to the low HER activity and weak E_B_[CO] [18]. Ag is a more cost-effective catalyst for selective reduction of CO_2_ to CO compared to Au, which is more than 70 times more expensive than Ag [19]. Zn processes the ability to form CO in CO_2_RR, which is an earth-abundant metal and much cheaper than Ag. Although Zn does not show satisfying activity and selectivity, it is one of the most promising non-previous metals for CO_2_RR in the CO-forming class [20]. Jaramillo and Nørskov et al. reported that the binding energy of adsorbed COOH (COOH*, E_B_[COOH]) is a key descriptor for CO generation efficiency in the CO_2_RR [21]. In the volcano plot of E_B_[COOH], Ag and Zn appear on the weak- and strong-binding legs of the plot respectively [12], therefore, Ag-Zn alloys would have a more appropriate E_B_[COOH] to improve the activity of CO_2_RR than either single metal (Ag or Zn) [12]. Taking advantage of the synergistic effect, Ag-Zn alloys may be able to increase the CO selectivity and activity of CO_2_RR while reducing the catalyst cost. 

High-throughput catalyst screening technique is a promising scientific method, which has been developed for the discovery and optimization of catalysts. Using this method can speed up the catalyst discovery process in exploring the complex catalyst composition space, such as alloying elements and surface molecules [22,23]. Applying the concept of high throughput in material preparation, samples with different element concentrations can be prepared directly at one time, significantly simplifying the sample preparation process, and providing convenience for the subsequent exploration of electrochemical properties.

In this work, to identify the relation between the CO_2_RR activity and the element concentrations as well as phase compositions of Ag-Zn alloys, we applied the concept of high throughput in material preparation and synthesized a series of Ag-Zn bimetallic alloys by composition gradient sputtering. The selectivity and activity of the CO_2_RR on catalysts with different element concentrations and phase compositions were compared for the efficient electrochemical production of CO.

## 2. Experimental

### 2.1. Catalysts Preparation

The Ag-Zn alloy catalysts were prepared by magnetron sputtering using carbon fiber paper (CFP) as the substrate. CFP was ultrasonically cleaned in ethanol for 5 min and dried at 60 °C in air. Subsequently, Ag and Zn were directly deposited onto the CFP using direct-current magnetron sputtering equipment (SKY Technology Development Co., Ltd., Shenyang, China) with a working distance of 6 cm. High-purity Ag and Zn (99.99 wt.%, Beijing Dream Material Technology Co., Ltd., Beijing, China) were used as the sputtering targets. The normal inclination angle between the target and CFP substrate was 20°, and the sample holder stood still to obtain samples with concentration gradients [24,25]. The sputtering was operated at room temperature for 1 h, with a power of 25 W for the Ag target and 35 W for the Zn target, and an Ar pressure of 1 Pa at a flow rate of 30 cm^3^ min^−1^. Similarly, pure Ag and Zn were also sputtered on CFP, respectively. The sputtering power was 25 W for Ag and 35 W for Zn at a rotating speed of 5 revolutions per minute for the substrate. Finally, we obtained uniform Ag and Zn films with a mass loading of around 0.7 mg cm^−2^, and Ag-Zn alloy films with different concentrations and a mass loading of about 1.4 mg cm^−2^.

### 2.2. Material Characterizations

X-ray diffraction (XRD) patterns were recorded by an XD-3 diffractometer equipped with Cu Kα radiation (Beijing Purkinje General Instrument Co., Ltd., Beijing, China). The microstructures and chemical compositions of the samples were characterized by a scanning electron microscope (SEM, COXEM EM-30, Daejeon, Korea) equipped with an energy dispersive X-ray (EDX) analyzer. 

### 2.3. Electrochemical Measurements

Electrochemical measurements were conducted on a CHI 760E potentiostat in an H-type cell with an ion-exchange membrane separator (Nafion 117 membrane, FuelCell Store, TX, America) inserted between the cathodic and anodic chambers. 0.5 M KHCO_3_ solution was used as the electrolyte in both the cathode and anode compartments. The sputtering samples, a Pt foil, and a double-junction saturated calomel electrode (SCE, saturated KCl, TIANJINAIDA R0232, Tianjin, China) were used as the working, counter, and reference electrodes, respectively. The potentials reported in this work were converted to be against the reversible hydrogen electrode (RHE) according to the following equation:(1)$$E_{\mathrm{RHE}}{=E}_{\mathrm{SCE}}+0.241+0.059\;\times \;\mathrm{PH}$$

A chronoamperometric electrolysis test was carried out at the working electrode with a geometric area of 1 cm^2^. Before the test, the fresh electrolyte solution was bubbled with high-purity CO_2_ for 15 min, and the test was under a continuous CO_2_ (99.9%) gas flow of 20 mL min^−1^. The gas-phase products from the gastight cathodic compartment were directly introduced into a gas chromatograph (GC, Lunan Instrument GC-7820, Jinan, China) which used high-purity Ar as the carrier gas for all compartments and analyzed using a thermal conductivity detector (TCD) and two flame ionization detectors (FID1 and FID2). The amount of CO gas was estimated from the FID1 data and H_2_ gas from the TCD data. An external standard method was adopted to quantify the products. The gas-phase products were injected into GC every 12 min during the CO_2_RR in the chronoamperometry measurement mode. 

## 3. Results and Discussion

### 3.1. Preparation and Characterization of the Ag-Zn Alloys as well as Pure Ag, Zn Films 

As illustrated in Figure 1a, the Ag-Zn alloy film was sputtered on CFP by a co-sputtering method. The macroscopic picture of the sputtering film in Figure 1b, shows a color gradient, indicating a corresponding concentration gradient in different regions. The sample appears black gray and bright silver on the side near the Zn target and the Ag target, respectively, and shows a color transition between the two regions. The concentration gradient distribution of the obtained Ag-Zn alloy film is shown in Figure 1c,d based on EDX analysis, and several respective results (e.g., Ag_10_Zn_90_, Ag_20_Zn_80_, Ag_30_Zn_70_, Ag_35_Zn_65_, Ag_40_Zn_60_, Ag_45_Zn_55_, Ag_55_Zn_45_, Ag_70_Zn_30,_ and Ag_85_Zn_15_) are shown in Appendix A. In addition, pure Ag and Zn films with a uniform composition were prepared by sputtering with the single respective metal target. The macroscopic view of the pure Zn and Ag films shows that the film color is consistent with the respective metal target (Appendix A). 

The morphologies and structures of the Ag-Zn alloy films were investigated by SEM. It obviously shows that the surface of pure Zn film is grainy (Figure 2a), while it is smooth and without obvious features for the pure Ag film (Figure 2b). As shown in Appendix A, with the Ag concentration increasing, the roughness of the film surface decreases and becomes smooth gradually. When the atomic percentage of Ag reaches 20%, the film becomes relatively smooth, and a further increase in Ag concentration shows little effect on the morphology. The Ag_45_Zn_55_ alloy shows similar surface morphology as that of pure Ag film as shown in Figure 2c. The EDX mappings (Figure 2d–f) show both Ag and Zn coat well on the fiber surface of the CFP.

The phase constitutions of the selected samples with different concentrations were characterized by XRD in Figure 3a. The characteristic peaks at 18.0°, 26.4°, and 54.5° are ascribed to CFP (Appendix A). The diffraction peaks of pure Ag film at 38.1°, 44.3°, 64.4°, 77.4°, and 81.5° are consistent with the standard Ag phase (PDF #65-2871). For the Ag_55_Zn_45_, Ag_70_Zn_30,_ and Ag_85_Zn_15_ catalysts, the diffraction peaks of Ag are observed to shift towards higher 2θ angles, indicating the formation of Ag(Zn) solid solution alloy associated with the lattice distortion/contraction (Figure 3b) [26]. As shown in Figure 3c,d, other catalysts with a smaller atomic ratio of Ag to Zn consist of the AgZn phase (Ag_45_Zn_55_, Ag_40_Zn_60,_ and Ag_35_Zn_65_), Ag_5_Zn_8_ phase (Ag_30_Zn_70_ and Ag_20_Zn_80_), AgZn_3_ phase (Ag_10_Zn_90_) and Zn phase (pure Zn catalyst) according to their diffraction peak positions. Therefore, a series of Ag-Zn alloy films with different compositions and phases were successfully fabricated through the high throughput co-sputtering method.

### 3.2. Electrochemical CO_2_RR Performance over Ag-Zn Alloys as well as Pure Ag, Zn Films in the H-Cell

The electrocatalytic performance of CO_2_RR was explored over the Ag-Zn alloys in CO_2_-saturated 0.5 M KHCO_3_ by using a gas-tight H-cell. For comparison, the electrochemical CO_2_RR performance of the pure Ag and Zn films was also evaluated under the same reaction conditions. Potentiostatic experiments were conducted at 7 different potentials in the range of −0.6 to −1.2 V vs. RHE. The faradaic efficiency of *x* (FE_x_) production and its partial current density (*j*_x_) was calculated as follows [12,27]:(2)$${\mathrm{FE}}_{x}\;\left(\%\right)=\frac{n_{x}\left(\mathrm{mol}\right)\;\cdot \;N\;\cdot \;F\;\left({C\;\mathrm{mol}}^{-1}\right)}{\;Q\;\left(C\right)}\;\times \;100=\frac{0.1315\;\cdot \;V\;\left(\frac{\mathrm{mL}}{\min}\right)\;\cdot {\;R}_{x}}{1000\;\cdot {\;i}_{\mathrm{total}}\;(\mathrm{mA})}\;\times \;100\;$$
(3)$$j_{x}\;\left({\mathrm{mA}\;\mathrm{cm}}^{-2}\right)=\frac{i_{x}\left(\mathrm{mA}\right)}{\;A\;\left({\mathrm{cm}}^{2}\right)}=\frac{{\;i}_{\mathrm{total}}\;(\mathrm{mA})\;\cdot {\;\mathrm{FE}}_{x}}{\;A\;\left({\mathrm{cm}}^{2}\right)}$$
where *n_x_* is the amount of product x (mol), *N* is the number of electrons transferred for product *x* formation (*N* is 2 for product CO and H_2_), *F* is the Faraday constant (96,485 C mol^−1^), *Q* is the charge passed to produce *x* (C), V is the actual flow rate during the test, *R_x_* is the concentration of product *x* from the cathode compartment (obtained from GC data), *i_x_* is the current value of product *x* (mA), and A is the geometric surface area (cm^2^). In general, the reduction mechanism of CO_2_ to CO is as follows [28]:(4)$${\mathrm{CO}}_{2}\left(g\right)+*+H^{+}\left(\mathrm{aq}\right)+e^{-}\;↔{\;\mathrm{COOH}}^{*}$$
(5)$${\;\mathrm{COOH}}^{*}+H^{+}\left(\mathrm{aq}\right)+e^{-}\;↔\;{\mathrm{CO}}^{*}+H_{2}O\left(l\right)$$
(6)$${\mathrm{CO}}^{*}\;↔\;\mathrm{CO}\left(g\right)+*$$
where * denotes an active site. A promising catalyst for selectively reducing CO_2_ to CO needs to appropriately immobilize COOH*, therefore, to facilitate the formation of CO* in the next elementary step (Equation (5)), and to bind CO weakly so that to release the active site easily (Equation (6)). That is, a relatively stronger binding of COOH* than CO* is necessary for efficient CO_2_RR [17,18]. 

The faradaic efficiency of CO (FE_CO_), H_2_ (FE_H_2__) as well as CO and H_2_ (FE_CO&H_2__) of CO_2_RR was plotted as a function of applied potentials in Figure 4. CO and H_2_ are the major products of the Ag-Zn alloy, indicating that the Ag-Zn alloy is a CO-forming catalyst, akin to Ag and Zn [29]. For all Ag-Zn film electrodes, H_2_ is the major product at less negative potentials due to the high overpotential required for CO_2_RR [18,30]. Whereas under the moderate potentials, CO is the major product. There is enough electrochemical driving force to reduce CO_2_ to CO and sufficient CO_2_ supply at the electrode interface so that the reaction rate of CO_2_ to CO exceeds that of HER generated from water [31]. At more negative potentials, the mass-transport limitations of CO_2_ may suppress CO yielding [18,28].

As shown in Figure 4a, different phase compositions and concentrations of the Ag-Zn alloy catalysts exhibit different trends of FE_CO_ at −0.6~−1.2 V vs. RHE. When the Ag content is high and Ag/Ag(Zn) solid solution phase is displayed, the maximum FE_CO_ appears in the range of −0.9~−1.0 V vs. RHE, compared to the range of −0.8~−0.9 V vs. RHE for other AgZn and Zn phases. For the Ag(Zn) phase catalysts, pure Ag catalyst shows the highest FE_CO_ over the entire potential range investigated (Figure 4a), and the addition of Zn to Ag decreases CO yielding (Appendix A), showing that Ag is more effective at reducing CO_2_ to CO than the Ag-Zn alloys or Zn. This trend can be explained by the changing of E_B_[CO] [17,28]. The volcano-type relation of the partial current density of CO as a function of E_B_[CO] shows that the catalyst surface with a weak E_B_[CO] can be limited by the activation of CO_2_ due to the instability of the COOH* intermediate, whereas surfaces with a strong E_B_[CO] are limited by highly stabilized CO* [17]. Both Ag and Zn have weak E_B_[CO] and belong to the category of surfaces limited by the activation of CO_2_. Ag is closer to the top of the volcano plot compared to Zn, and better for CO_2_ activation, which explains why Ag is more effective for CO formation [31,32]. When the alloy films consist of the AgZn phase, FE_CO_ decreases with the increase of Ag concentration at the intermediate potentials, while has little change at low and high potentials (Appendix A). Pure Zn catalyst shows the lowest current efficiency for CO production, and the small amount addition of Ag (10 at.%) significantly improves the FE_CO_ by ~100% at −0.6~−0.7 V vs. RHE (Appendix A). At higher potentials, the AgZn_3_ phase catalyst shows the closest FE_CO_ to pure Ag catalyst, but with the potential decreasing, it is consistent with the pure Zn. The CO yielding of the Ag_5_Zn_8_ catalyst is higher than that of pure Zn and increases with the increase of Ag concentration at lower potentials. This indicates that the AgZn_3_ phase catalyst is favorable for CO production at higher potentials, while the Ag_5_Zn_8_ phase catalyst is favorable at lower potentials. The scenario of increased CO selectivity of the Ag-Zn alloys compared to Zn at intermediate potentials may originate from the electron transfer from Ag to Zn [33], due to the lower work function of Ag (4.26 eV) than Zn (4.33 eV) [34,35,36]. The transferred electrons are found to fill the bonding states of localized d-orbitals and exhibit a conspicuous upshift toward the Fermi level, which enhances the adsorption of the intermediate COOH*, resulting in a higher catalytic activity for CO production compared to pure Zn catalysts [35,37]. 

Figure 4b shows the changes of FE_H_2__ with potentials, which are basically the same for all Ag-Zn film electrodes. FE_H_2__ decreases rapidly with the potential decreasing, and tends to flatten or even slightly rebound at lower potentials. This indicates that water decomposition is more favorable at less negative potentials, while more active sites are occupied by CO_2_ and related intermediates at more negative potentials. Pure Ag, Ag_5_Zn_8_ (Ag_30_Zn_70_ and Ag_20_Zn_80_ catalysts), AgZn_3_ (Ag_10_Zn_90_ catalyst), and pure Zn phase catalysts exhibit lower FE_H_2__, whereas that of Ag(Zn) solid solution and AgZn phase catalysts is higher (Appendix A), indicating that the former can inhibit HER better. 

It can be seen from Figure 4c that the combined faradaic efficiency of CO and H_2_ can reach over 80% at the less negative potential range, and almost no other products are produced. This is due to the weak E_B_[CO] at lower overpotential, leading to the rapid desorption of CO from the surface once generated, and therefore, it is difficult to carry out further reduction [31]. In addition, other complex products require multiple proton-electron transfer steps and a larger driving force (more negative potentials) [18]. The FE_CO&H_2__ for Ag/Ag(Zn) solid solution alloy catalysts decreases by 20–30% at a more negative potential (Appendix A), that of AgZn phase catalysts decreases by about 30% (Appendix A), and that of Ag_5_Zn_8_, AgZn_3_ as well as pure Zn phase catalysts decreases by about 40% (Appendix A). This is ascribed to the formation of other liquid products, demonstrating an unfavorable competition between the further reduction of CO* and desorption of CO [18,31]. An explanation is that a higher overpotential provides a higher driving force and improves the electron transfer, which enhances CO* electroreduction, competing with the non-faradaic step of CO desorption [31].

Figure 5a presents the overall electrode activity of the Ag-Zn alloys as well as pure Zn and Ag as a function of the applied potentials. For all electrodes, a well-defined exponential increase in the overall activity is observed with decreasing potentials. With the addition of Zn in Ag, the total current densities (*j*_total_) increase for the Ag/Ag(Zn) solid solution phase catalysts at all potentials (Appendix A). The catalysts with other phases exhibit a similar level of *j*_total_, which is higher than that of the pure Ag catalyst (Appendix A). Notably, the AgZn_3_ phase catalyst (Ag_10_Zn_90_) exhibits an especially high *j*_total_. This indicates that Ag-Zn alloying will improve the activity of the catalysts, and the AgZn_3_ phase catalyst has the highest activity.

The partial current density towards specific CO_2_RR products is another important figure of merit which is useful in comparing CO_2_RR catalysts. Because the partial current density is directly proportional to the turnover frequency (TOF) of a certain product, the presentation of the data is useful in obtaining insights into the kinetics and mechanisms of the reduction reactions [18]. The partial current densities of CO (*j*_CO_) and H_2_ (*j*_H_2__) are presented in Figure 5b,c in the form of a semi-log Tafel plot, respectively. The observed trend of *j*_CO_ is similar for all catalysts (Figure 5b and Appendix A). In the less negative potential region, the TOF of CO formation increases with the decrease of potentials, whereas in the region of lower potentials, this growth of TOF gradually slows down and reaches a plateau, which is assigned to a CO_2_-mass-transport limitation in aqueous solution due to its low solubility at atmospheric pressure [31,38], consistent with FE_CO_ in Figure 4a. To obtain further insights into these catalysts, their Tafel plots were investigated and presented in Appendix A. A small Tafel slope is beneficial in practical applications because it will lead to a much faster increase in the increment of CO_2_ reduction rate with increasing overpotential [13]. For the catalysts with the Ag/Ag(Zn) solid solution phase, *j*_CO_ increases as a function of overpotential with a slope of 233, 221, 188, and 187 mV dec^−1^ for the Ag, Ag_85_Zn_15_, Ag_70_Zn_30,_ and Ag_55_Zn_45_ catalysts, respectively in the lower overpotential region (Appendix A). It is observed that the Tafel slope decreases gradually with a higher Zn content, and when the Ag-Zn alloy behaves as the AgZn phase, the Ag_40_Zn_60_ catalyst shows the lowest Tafel slope which is only 165 mV dec^−1^ (Appendix A). While for the Ag_5_Zn_8_ and AgZn_3_ phase catalysts, the Tafel slope is even larger than that of the pure Zn (Appendix A). These results indicate that the alloying of Ag with Zn can improve the reaction kinetics of CO_2_RR, and the enhanced degree depends on the phase composition and element concentration. 

As shown in Figure 5c, *j*_H_2__ of the catalysts with different phase compositions forms transient platforms in different potential regions. The platform of pure Ag and AgZn phase catalysts appears between −0.7~−0.8 V and −0.9~−1.1 V vs. RHE, respectively (Appendix A), and that of Ag_5_Zn_8_, AgZn_3_ and Zn phase catalysts appears between −0.7~−0.9 V vs. RHE (Appendix A). This reflects the observed shift in selectivity from hydrogen to CO in this platform region, as discussed earlier [18]. In the more negative potential region, an increase in the TOF of H_2_ is observed, which explains the slight rebound of H_2_ and the production of other products in this potential region.

## 4. Conclusions

In summary, a series of Ag-Zn thin alloy film catalysts with different concentrations were prepared by a simple co-sputtering method, and the effect of different element concentrations and phase compositions on the activity and selectivity of CO_2_RR was analyzed. It can be seen from the results that different phase compositions have affected the trend of FE_CO_ and the selectivity from hydrogen to CO with varying potentials. Pure Ag catalyst exhibits the best CO_2_ selectivity, and the alloying with Zn does not significantly increase the activity of CO_2_RR for the Ag/Ag(Zn) phase catalysts. The Ag_5_Zn_8_ and AgZn_3_ phase catalysts show higher CO selectivity than pure Zn due to the synergistic effect of the Ag-Zn alloy. The alloying of Ag with Zn could improve the activity of the catalysts and the reaction kinetics of CO_2_RR, and the AgZn_3_ phase catalyst has the highest activity. The present work could provide guidelines for the design of Ag-Zn alloy catalysts for efficient CO_2_RR. 

## Figures and Tables

**Figure 1 materials-15-06892-f001:**
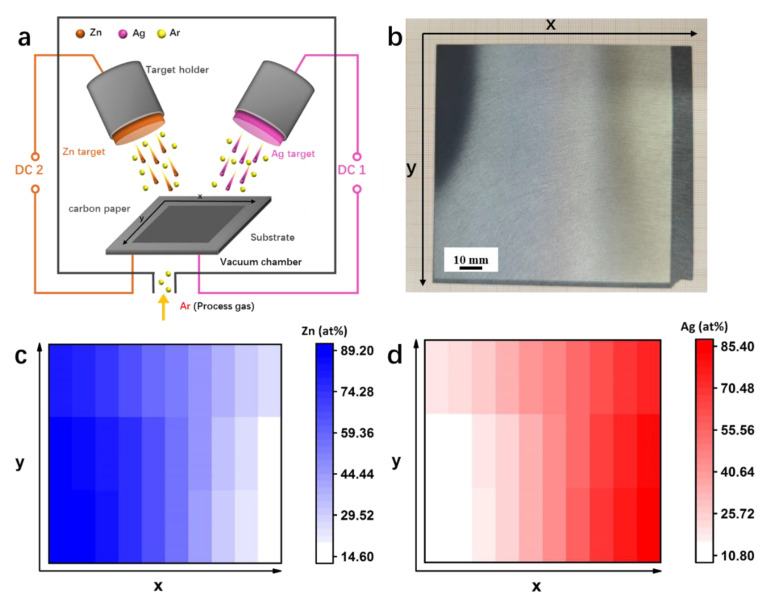
(**a**) Schematical illustration showing the preparation process of the Ag-Zn film catalysts by co-sputtering. (**b**) Macroscopic view of the Ag-Zn film after co-sputtering. (**c**,**d**) Concentration gradient distribution of (**c**) Zn and (**d**) Ag elements.

**Figure 2 materials-15-06892-f002:**
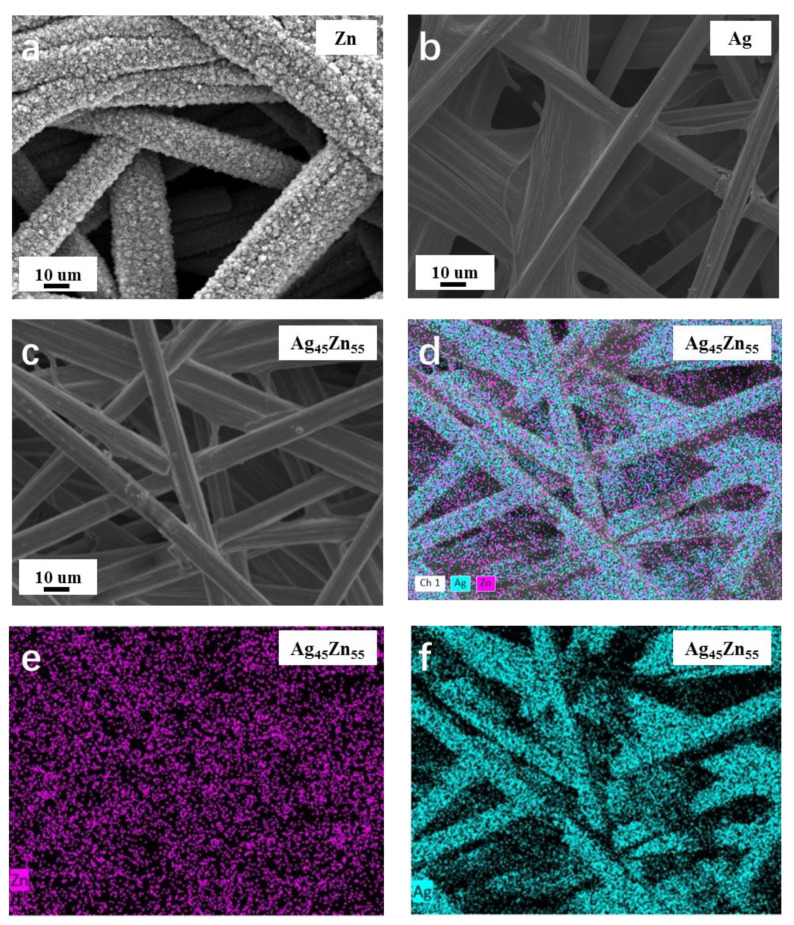
(**a**–**c**) SEM images of the (**a**) Zn, (**b**) Ag, and (**c**) Ag_45_Zn_55_ films. (**d**–**f**) EDX mapping images of the Ag_45_Zn_55_ film.

**Figure 3 materials-15-06892-f003:**
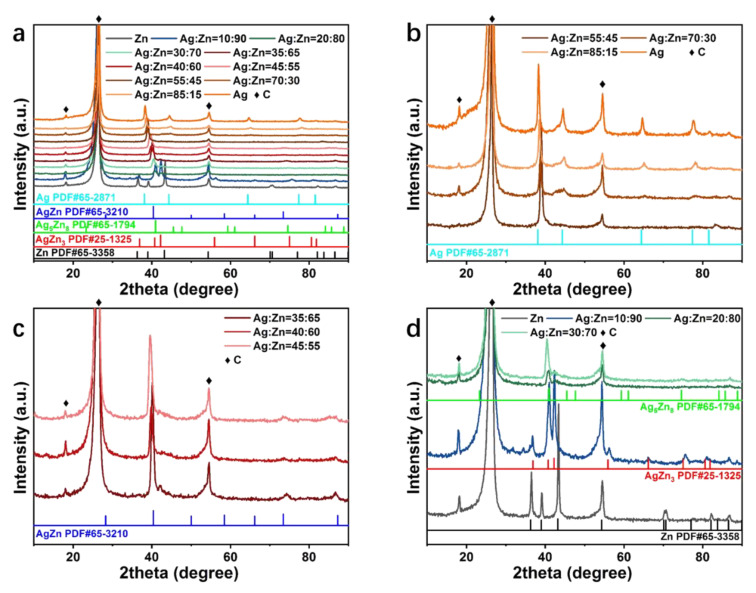
(**a**) XRD patterns of the pure Zn, pure Ag, and Ag-Zn alloy catalysts with different concentrations. (**b**–**d**) XRD patterns of the (**b**) Ag, Ag_55_Zn_45_, Ag_70_Zn_30_, Ag_85_Zn_15_, (**c**) Ag_35_Zn_65_, Ag_40_Zn_60_, Ag_45_Zn_55_, and (**d**) Ag_10_Zn_90_, Ag_20_Zn_80_, Ag_30_Zn_70_, and Zn samples.

**Figure 4 materials-15-06892-f004:**
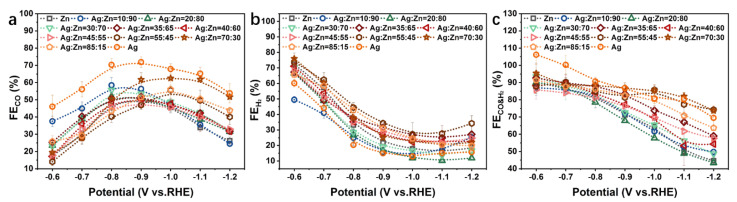
(**a**) FE_CO_, (**b**) FE_H_2,__ and (**c**) FE_CO&H_2__ for the pure Zn, pure Ag, and Ag-Zn alloy catalysts with different element concentrations.

**Figure 5 materials-15-06892-f005:**
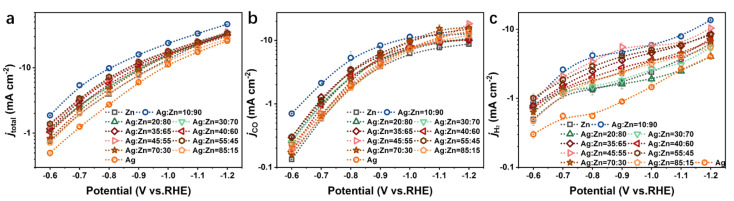
(**a**) *j*_total_, (**b**) *j*_CO,_ and (**c**) *j*_H_2__ for the pure Zn, pure Ag, and Ag-Zn alloy catalysts with different element concentrations.

## Data Availability

All data are available within the paper and its Appendix A files or from the corresponding authors upon request.

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
