# Peer review of "High Throughput Preparation of Ag-Zn Alloy Thin Films for the Electrocatalytic Reduction of CO2 to CO"

_materials, 2022, doi:10.3390/ma15196892_

Round 1

Reviewer 1 Report

The manuscript from Dr. Zhang and co-workers reported using magnetron co-sputtering method for preparation of gradient Ag-Zn alloy thin films in different composition for the electrocatalytic reduction of CO2 to CO testing. I should note that this manuscript clearly described and summarized to support the authors’ assertions.  Finally, I recommend acceptance of the manuscript after minor revision based on the following comments:

1.      Abstract: ‘The synthesis of cost-effective and high-performance electrocatalysts for CO2 reduction’ is not much related to the title.

2.      Abstract: Need to give the full name when first used the abbreviation of ‘CO2RR’.

3.      Page 1 : the diagram reaction is not balance for CO2 to CO.

4.      Figure 1b: give the scale bar for the diagram.

5.      Introduction, 3rd paragraph: What is CO*? should explain ‘*’ here not later in the manuscript.

6.      Experimental: How do you know  the Ag and Zn films are uniform?

7.      Page 7:’ * denotes an active site’ where is this active site?

Author Response

Point 1: Abstract: ‘The synthesis of cost-effective and high-performance electrocatalysts for CO2 reduction’ is not much related to the title.

Response 1: Thank you for your valuable advice. This sentence has been substituted by “Ag-Zn alloys are identified as high active and selective electrocatalysts for CO2 reduction, while how the phase composition of the alloy affects the catalytic performances has not been systematically studied yet.” and highlighted by “Track Changes” function in the revised manuscript.

Point 2: Abstract: Need to give the full name when first used the abbreviation of ‘CO2RR’.

Response 2: Thank you for your valuable advices. We have given the full name of ‘CO2RR’ as ‘CO2 reduction reaction’ when first mentioned in the abstract.

Point 3: Page 1: the diagram reaction is not balance for CO2 to CO.

Response 3: Thank you for your valuable advices. We have balanced the chemical equation of CO2 electrocatalytic reduction (CO2 + 2H+ +2e- → CO +H2O)  in the graphic abstract.

Point 4: Figure 1b: give the scale bar for the diagram.

Response 4: Thank you for your valuable advices. We have added the scale bar in Fig. 1b.

Point 5: Introduction, 3rd paragraph: What is CO*? should explain ‘*’ here not later in the manuscript.

Response 5: Thank you for your valuable advices. ‘*’ refers to the active sites on the catalyst surface and ‘CO*’ is denoted as ‘adsorbed CO’, and the corresponding explanation has been made in the introduction.

Point 6: Experimental: How do you know the Ag and Zn films are uniform?

Response 6: The Ag and Zn films were prepared by magnetron sputtering and the substrate was rotated at a speed of 5 revolutions per minute during sputtering, which ensured the synthesized Ag and Zn films are uniform.

Point 7: Page 7: ‘* denotes an active site’ where is this active site?

Response 7: The active sites refer to the metal sites on the catalyst surface, which are exposed to the CO2 dissolved in the electrolyte solution, so that to adsorb and convert CO2.

Reviewer 2 Report

Manuscript ID: materials-1925096

Referee Comments:                                                                                      Date: 09/16/2022

This manuscript by Jiameng Sun et al., entitled as “High throughput preparation of Ag-Zn alloy thin films for the electrocatalytic reduction of CO2 to CO” presents a series of Ag-Zn alloy catalysts films by magnetron co-sputtering grown on carbon fiber substrates as an electrocatalysts for CO2 reduction in an aqueous KHCO3 electrolyte. The as-synthesized films have multiple phases of Ag-Zn alloys, among which Ag5Zn8 and AgZn3 catalysts shows a relatively higher CO selectivity. Authors further observed that AgZn3 catalyst shows the highest activity for CO2RR.

The idea is interesting and could be useful for real time energy applications. However, there are some issues in this study and its presentation. The manuscript can be considered for publication in “materials” after revision. The detailed comments are listed as follows:

Abstract and Introduction:

1.     Why the AgZn3 and Ag5Zn8 phase catalyst has higher selectivity for CO production? What physical and chemical properties make these phases unique over other phases?

2.     Is it possible to control individual phase by magnetron co-sputtering sputtering throughout the CFP?

3.     What is the synergistic effect of AgZn3 and Ag5Zn8 phase catalysts shows higher CO selectivity than pure Zn?

4.     Could you provide the chemical role in between the electrode electrolyte interface affecting its physical properties as discussed in the following article:

https://doi.org/10.1002/aesr.202100137

5.     How do you compare the results of this work as compared to reported literature? For instance:

https://doi.org/10.1002/ente.201700087

Author Response

Point 1: Why the AgZn3 and Ag5Zn8 phase catalyst has higher selectivity for CO production? What physical and chemical properties make these phases unique over other phases?

Response 1: As shown in Fig. S5c (the original Fig. 4d), the AgZn3 phase catalyst is favorable for CO production at higher potentials, while the Ag5Zn8 phase catalyst is favorable at lower potentials, compared to pure Zn. This may be ascribed to the electron transfer from Ag to Zn in AgZn3 and Ag5Zn8 phase catalysts [1], which fills the bonding states of localized d-orbitals and exhibits a conspicuous upshift toward the Fermi level, therefore, enhances adsorption of the intermediate COOH*, resulting in a higher catalytic activity for CO production compared to other phase catalysts [2, 3]. A further detailed study will focus on the relation of the structure/property and performances of the AgZn3 and Ag5Zn8 phase catalysts.

Point 2: Is it possible to control individual phase by magnetron co-sputtering sputtering throughout the CFP?

Response 2: Ag-Zn alloy films with specific composition can form individual phase. For example, as shown in Fig.3, the Ag-Zn alloy film is merely composed of AgZn phase for the Ag:Zn ratio close to 1:1 (at.%). It can be inferred that a sample with a certain composition obtained by controlling the sputtering power of Ag and Zn, the Ag-Zn films with individual phase can be obtained throughout the CFP which is rotated during the sputtering.

Point 3: What is the synergistic effect of AgZn3 and Ag5Zn8 phase catalysts shows higher CO selectivity than pure Zn?

Response 3: In the volcano plot of EB[COOH], Ag and Zn appear on the weak- and strong-binding legs of the plot respectively [4]. The combination of Ag and Zn in the AgZn3 and Ag5Zn8 phase catalysts can provide adequate EB[COOH] for efficient CO formation compared to pure Zn catalyst [5]. And the related part is added into the revised manuscript.

Point 4: Could you provide the chemical role in between the electrode electrolyte interface affecting its physical properties as discussed in the following article:

https://doi.org/10.1002/aesr.202100137

Response 4: We provide a comparison about how the electrochemical test affects the bimetallic catalyst based on Ag and Zn after a detail literature survey. For the nanoporous Ag electrode, the roughness factor of the Ag nanoparticles (i.e., ECSA) increased by 20.2% compared with that of the initial state after 1 h of CO2RR [6]. In addition, the increase of surface roughness of Ag and Zn foils can also be seen by SEM images during the electrolysis experiment, which is attributed to native oxide reduction and increases the number of active sites [7-9]. But for some Ag-Zn alloys, the morphology of the Ag-Zn alloy foils did not change significantly after electrochemical investigations [8, 10]. An increase in Zn at% is observed on Ag-Zn alloy foils surfaces, which may be due to the strong interaction between Zn atoms on the electrode surface and carbonate and bicarbonate ions in the electrolyte, making the similar surface of Ag-Zn alloy and Zn, resulting in the similar properties [8]. Thanks for your nice advice. We will consider this scenario in the following work.

Point 5: How do you compare the results of this work as compared to reported literature? For instance:

https://doi.org/10.1002/ente.201700087

Response 5: In accordance with the study of Toru Hatsukade et al. [8], Ag is more effective at reducing CO2 to CO than the Ag-Zn alloys or Zn. The most advanced AgZn-based catalysts reported show superior CO selectivity over single metals, especially pure Zn catalysts, such as 2 nm Ag layer deposited catalyst on Zn (84%) [11], CP/PPy/Zn/Ag0.18 (70%) [5], anodized Zn/C/Ag (86%) [12] and superfine Ag nanoparticle decorated Zn nanoplates (84%) [13]. This is consistent with the conclusion that the enhanced electrochemical reduction activity and selectivity of AgZn alloy are attributed to the synergistic interaction between Ag and Zn in this work.

References:

  1. Hansen H A, Varley J B, Peterson A A, Norskov J K. Understanding trends in the electrocatalytic activity of metals and enzymes for CO2 reduction to CO. Journal of Physical Chemistry Letters 2013; 4 (3): 388-392.
  2. Zhao Z, Lu G. Computational screening of near-surface alloys for CO2 electroreduction. ACS Catalysis 2018; 8 (5): 3885-3894.
  3. Zhang Z, Wen G, Luo D, Ren B, Zhu Y, Gao R, Dou H, Sun G, Feng M, Bai Z, Yu A, Chen Z. “Two Ships in a Bottle” design for Zn-Ag-O catalyst enabling selective and long-lasting CO2 electroreduction. Journal of the American Chemical Society 2021; 143 (18): 6855-6864.
  4. Feaster J T, Shi C, Cave E R, Hatsukade T, Abram D N, Kuhl K P, Hahn C, Nørskov J K, Jaramillo T F. Understanding selectivity for the electrochemical reduction of carbon dioxide to formic acid and carbon monoxide on metal electrodes. ACS Catalysis 2017; 7 (7): 4822-4827.
  5. Jo A, Kim S, Park H, Park H-Y, Hyun Jang J, Park H S. Enhanced electrochemical conversion of CO2 to CO at bimetallic Ag-Zn catalysts formed on polypyrrole-coated electrode. Journal of Catalysis 2021; 393: 92-99.
  6. Yun H, Kim J, Choi W, Han M H, Park J H, Oh H-s, Won D H, Kwak K, Hwang Y J. Understanding morphological degradation of Ag nanoparticle during electrochemical CO2 reduction reaction by identical location observation. Electrochimica Acta 2021; 371: 137795.
  7. Hatsukade T, Kuhl K P, Cave E R, Abram D N, Jaramillo T F. Insights into the electrocatalytic reduction of CO2 on metallic silver surfaces. Physical Chemistry Chemical Physics 2014; 16 (27): 13814-13819.
  8. Hatsukade T, Kuhl K P, Cave E R, Abram D N, Feaster J T, Jongerius A L, Hahn C, Jaramillo T F. Carbon dioxide electroreduction using a silver-zinc alloy. Energy Technology 2017; 5 (6): 955-961.
  9. Parmar S, Das T, Ray B, Debnath B, Gosavi S, Shanker G S, Datar S, Chakraborty S, Ogale S. N, H Dual-doped black anatase TiO2 thin films toward significant self-activation in electrocatalytic hydrogen evolution reaction in alkaline media. Advanced Energy and Sustainability Research 2021; 3 (1): 2100137.
  10. Low Q H, Loo N W X, Calle-Vallejo F, Yeo B S. Enhanced electroreduction of carbon dioxide to methanol using zinc dendrites pulse-deposited on silver foam. Angewandte Chemie International Edtion 2019; 58 (8): 2256-2260.
  11. Guo W, Shim K, Kim Y-T. Ag layer deposited on Zn by physical vapor deposition with enhanced CO selectivity for electrochemical CO2 reduction. Applied Surface Science 2020; 526: 146651.
  12. Gao Y, Li F, Zhou P, Wang Z, Zheng Z, Wang P, Liu Y, Dai Y, Whangbo M-H, Huang B. Enhanced selectivity and activity for electrocatalytic reduction of CO2 to CO on an anodized Zn/carbon/Ag electrode. Journal of Materials Chemistry A 2019; 7 (28): 16685-16689.
  13. Yu Q, Meng X, Shi L, Liu H, Ye J. Superfine Ag nanoparticle decorated Zn nanoplates for the active and selective electrocatalytic reduction of CO2 to CO. Chemical communications 2016; 52 (98): 14105-14108.

Reviewer 3 Report

Comments and recommendations:

The authors in this paper have fabricated a series of Ag-Zn alloy catalysts by magnetron co-sputtering, and further explored their activity and selectivity towards CO2 electroreduction.This paper has some merits that could be of interest to the reader, however, a minor revision is needed in order to be suitable for publication.

Comments and suggestions are attached.Comments and suggestions are listed as follows:

1.     The last paragraph in the abstract (Despite failure to improve the CO selectivity, Zn alloying improves the catalytic activity and reaction kinetics of CO2RR, and the AgZn3 catalyst shows the highest activity. This work is inspiring to explore Ag-Zn alloy catalysts with promising CO2RR properties ) is hard to follow, please rewrite with emphasising the potential applications and novelty of the work.

2.     In the Catalysts preparation section , references needed to be included.. 

3.     The paper needs to be proofread, font size and style needs to be consistent throughout..

4.     In page 4, you mentioned “as the concentration increasing, the roughness of the film surface decreases and becomes smooth gradually “, could you quantify the roughness of the surface? Is there any advantages when the surface become rougher ?

5.     Can you move some of the Figs from Fig 4 to 9 to the SP, and keep only the main one in the main paper 

6.     In the conclusion you have indicated “ The alloying of Ag with Zn could improve the activity of the catalysts and the reaction kinetics of CO2RR, and the AgZn3 phase catalyst has the highest activity “, is there a specific reason for such observation? Has this been reported before in literature?

Author Response

Point 1: The last paragraph in the abstract (Despite failure to improve the CO selectivity, Zn alloying improves the catalytic activity and reaction kinetics of CO2RR, and the AgZn3 catalyst shows the highest activity. This work is inspiring to explore Ag-Zn alloy catalysts with promising CO2RR properties) is hard to follow, please rewrite with emphasizing the potential applications and novelty of the work.

Response 1: Thank you for your valuable advice. We replaced this paragraph with “Zn alloying improves the catalytic activity and reaction kinetics of CO2RR, and the AgZn3 catalyst shows the highest apparent electrocatalytic activity. This work found that the activity and selectivity of CO2RR are highly dependent on the element concentrations and phase compositions, which is inspiring to explore Ag-Zn alloy catalysts with promising CO2RR properties.” and highlighted by “Track Changes” function in the revised manuscript.

Point 2: In the Catalysts preparation section, references needed to be included.

Response 2: Thank you for your valuable advice. We have added corresponding references in the ‘Catalysts preparation’ section.

Point 3: The paper needs to be proofread, font size and style needs to be consistent throughout.

Response 3: Thank you for your valuable advice. We have proofread and revised the manuscript for consistency in font size and style.

Point 4: In page 4, you mentioned “as the concentration increasing, the roughness of the film surface decreases and becomes smooth gradually”, could you quantify the roughness of the surface? Is there any advantages when the surface become rougher?

Response 4: The roughness (R) can be characterized by electrochemical surface area (ECSA) and geometric surface area (A) ( R = ECSA / A ), and ECSA can be determined by analyzing the double layer capacitances (Cdl). While in this work, we focused on the influence of element concentrations and phase compositions of Ag-Zn alloys and didn’t spend much efforts on the surface roughness of the catalyst. Generally speaking, for a certain catalyst with the same intrinsic activity, the enhancement of surface roughness, that is to increase the electrochemical surface area, can expose more active sites to the reactant, therefore to increase the conversion efficiency than those with smooth surface.

Point 5: Can you move some of the Figs from Fig 4 to 9 to the SP, and keep only the main one in the main paper.

Response 5: Thank you for your valuable advice. We merged the Fig. 4a-9a into Fig. 4-5, and moved the other figures to the SP (Fig. S5-9 and Fig. S11).

Point 6: In the conclusion you have indicated “The alloying of Ag with Zn could improve the activity of the catalysts and the reaction kinetics of CO2RR, and the AgZn3 phase catalyst has the highest activity”, is there a specific reason for such observation? Has this been reported before in literature?

Response 6: The Ag-Zn alloy catalysts give a higher total current density than that of pure Ag and Zn catalyst (Fig. 4b, the original Fig.5a), and the AgZn3 phase catalyst (Ag10Zn90) exhibits the highest jtotal, illustrating that the alloying of Ag with Zn could improve the activity of the catalysts, and the AgZn3 phase catalyst has the highest activity. The main reason for this scenario is the synergistic effect between Ag and Zn, leading to an enhancement of the intrinsic activity of the catalysts compared with the single metal. In addition, the surface of AgZn3 phase catalyst (Ag10Zn90) is rougher than that of other Ag-Zn alloy catalysts (Fig. S3), therefore exposing more active sites and leading to a higher activity. It has also been stated in the published literature that the bimetallic Ag-Zn catalyst shows a higher total activity in comparison to the single metal, e.g. CP/PPy/Zn/Ag [1], Ag layer deposited catalyst on Zn [2] and anodized Zn/C/Ag [3], indicating that the synergistic effect between Ag and Zn could improve the electrocatalytic activity of CO2RR.

References:

  1. Jo A, Kim S, Park H, Park H-Y, Hyun Jang J, Park H S. Enhanced electrochemical conversion of CO2 to CO at bimetallic Ag-Zn catalysts formed on polypyrrole-coated electrode. Journal of Catalysis 2021; 393: 92-99.
  2. Guo W, Shim K, Kim Y-T. Ag layer deposited on Zn by physical vapor deposition with enhanced CO selectivity for electrochemical CO2 reduction. Applied Surface Science 2020; 526: 146651.
  3. Gao Y, Li F, Zhou P, Wang Z, Zheng Z, Wang P, Liu Y, Dai Y, Whangbo M-H, Huang B. Enhanced selectivity and activity for electrocatalytic reduction of CO2 to CO on an anodized Zn/carbon/Ag electrode. Journal of Materials Chemistry A 2019; 7 (28): 16685-16689.

Round 2

Reviewer 2 Report

The revision is satisfactory. The manuscript can be published as it is.